# Integrative Meta-Analysis of Huntington’s Disease Transcriptome Landscape

**DOI:** 10.3390/genes13122385

**Published:** 2022-12-16

**Authors:** Nela Pragathi Sneha, S. Akila Parvathy Dharshini, Y.-H. Taguchi, M. Michael Gromiha

**Affiliations:** 1Department of Biotechnology, Bhupat and Jyoti Mehta School of Biosciences, Indian Institute of Technology Madras, Chennai 600036, Tamilnadu, India; 2Department of Physics, Chuo University, Kasuga, Bunkyo-ku, Tokyo 112-8551, Japan

**Keywords:** Huntington’s disease, Brodmann Area 9, tissue-specific network analysis, variant effect, function interaction network

## Abstract

Huntington’s disease (HD) is a neurodegenerative disorder with autosomal dominant inheritance caused by glutamine expansion in the Huntingtin gene (HTT). Striatal projection neurons (SPNs) in HD are more vulnerable to cell death. The executive striatal population is directly connected with the Brodmann Area (BA9), which is mainly involved in motor functions. Analyzing the disease samples from BA9 from the SRA database provides insights related to neuron degeneration, which helps to identify a promising therapeutic strategy. Most gene expression studies examine the changes in expression and associated biological functions. In this study, we elucidate the relationship between variants and their effect on gene/downstream transcript expression. We computed gene and transcript abundance and identified variants from RNA-seq data using various pipelines. We predicted the effect of genome-wide association studies (GWAS)/novel variants on regulatory functions. We found that many variants affect the histone acetylation pattern in HD, thereby perturbing the transcription factor networks. Interestingly, some variants affect miRNA binding as well as their downstream gene expression. Tissue-specific network analysis showed that mitochondrial, neuroinflammation, vasculature, and angiogenesis-related genes are disrupted in HD. From this integrative omics analysis, we propose that abnormal neuroinflammation acts as a two-edged sword that indirectly affects the vasculature and associated energy metabolism. Rehabilitation of blood-brain barrier functionality and energy metabolism may secure the neuron from cell death.

## 1. Introduction

Huntington’s disease (HD) is a neurodegenerative disorder with a genetic inherent caused by glutamine expansion in the Huntingtin gene (HTT) on chromosome 4 [1,2,3,4]. Usually, the normal population has an average of 17–20 CAG (Glutamine) repeats in the HTT gene, but 40 or more CAG repeats are present in HD. HD patients show typical symptoms such as motor impairment, cognitive defects, chorea, and behavioral changes. Tetrabenazine and deep brain stimulation in pallidum reduce motor-related symptoms. However, in some HD patients, the cognitive deficit can affect memory association and leads to dementia [5]. Most of the current treatments are prescribed based on the symptoms. Selective vulnerability (in the striatum) is the predominant factor in the initial stages of the disease. The striatum is the nucleus of the basal ganglia and is involved in voluntary movement control, cognition, and decision-making. The striatum has major connectivity arbors to the frontal cortex, and in HD, prominent cell loss occurs in the cerebral cortex and striatum, affecting the cortico-striatal connection [6]. The cause of vulnerable neuronal death is unknown [7]. Expression studies can be performed on the striatum directly, but the highly degenerated tissue makes the interpretation difficult. Brodmann Area 9 (BA9) is the Dorsolateral Prefrontal Cortex (DLPFC), primarily connected with the executive subregion of the striatum and involved in motor planning and organization, regulation of intellectual function, and action. Analyzing the gene and transcript expression related to this area may aid in understanding the neuronal vulnerability in HD.

HD gene expression studies showed that heat shock proteins (HSP1A, HSP1B, and HSPB1) are upregulated in HD and havoc in the transcription factor networks affects the downstream gene expression [8]. Human iPSC cell studies demonstrated that downregulation in glutamate and GABA signaling, as well as destruction in calcium homeostasis, leads to apoptosis [9,10,11]. Lim et al. [12] reported that HD iPSC-derived brain microvascular endothelial cells have intrinsic abnormalities in angiogenesis and blood-brain barrier properties, as well as their signaling pathways. Expression studies from the putamen and cingulate cortex revealed that the dysregulation in the mitochondrial networks and astrocyte function collaterally induces inflammation [13,14]. Seredenina et al. [15] proposed that mutant HTT indirectly affects transcriptional regulation and miRNA binding that consecutively dysregulate the downstream gene expression. Epitranscriptome studies showed that repressing H3K4me3 promotors in human brain cells is beneficial in controlling the abnormal histone acetylation pattern [16,17]. Yildirium et al. [18] observed that aberrant transcriptional changes in H3K27 acetylation lead to changes in downstream expression patterns and an imbalance in histone acetylation and methylation patterns associated with HD. Dalamah et al. [13] performed single-nuclei experiments enriched for astrocyte dynamic states and identified four transcriptionally distinct states of astrocytes, which dysregulates calcium ion channel, synaptic transmission and oxidative phosphorylation.

Recent investigations discussed earlier focused on gene expression change and associated biological functional impact. These studies missed considering the transcript expression changes in HD conditions and how potential GWAS variants affect the expression by modulating regulatory elements and epigenetic motifs. In this work, we quantified the gene/transcript abundance and transcript proportions from BA9 control/HD samples to identify the interlink association between variants gene and transcript expression. In addition, we elucidated the variants from bulk RNA seq and compared them with genome-wide association studies with various neurodegenerative disorders [19,20,21,22]. Further, we predicted the effect of variants on regulatory functions such as miRNA, transcription factor (TF) binding, and histone acetylation and methylation pattern. 

We found that most variants affect the histone acetylation pattern and are upregulated in HD. This shows an imbalance in the regulatory network, which hampers the acetylation and methylation pattern. Predominant variants overlapped with transcripts and potentially changed their proportions. In addition, we identified variants affecting miRNA binding and associated target gene expression profiles. Differential gene expression studies revealed that mitochondrial, inflammatory-responsive genes are dysregulated at grade 3. Vasculature and angiogenesis-related functional genes are impacted at grade 4. From this study, we propose that neuroinflammation act as a bidirectional sword; it affects the endothelial cell and disrupts the BBB homeostasis. On the other hand, loss of vasculature affects energy metabolism and neurovascular coupling. Restoring vasculature and BBB homeostasis and reducing inflammatory signals may save the neuronal population from cell death.

## 2. Methods

### 2.1. Dataset and Pre-Processing

The transcriptomic data of the dorsolateral prefrontal cortex (Brodmann Area 9) from the tissue of HD patients and control samples were obtained from the SRA database (SRP051844) [23]. These are human post-mortem samples, and the age of death for HD patients varies from 40 to 75 and 36 to 106 for healthy controls. The HD samples were grouped according to their severity of neuropathological involvement, i.e., Vonsattel grading [24]. Vonsattel grading varies from 0 to 4 based on neuronal loss. Grade 0 has no observable abnormalities, and grade 4 has a 95% loss of neurons. The samples belong to grades 3 and grade 4. The RNA Seq data is in FASTQ format, and it is pre-processed to check the overall sequence quality. The transcriptomic data is prone to read errors, overrepresented sequences, or any experimental artifacts. It is essential to clean the raw data to avoid erroneous results in subsequent analyses, and processing of the raw data is performed using NGSQCToolkit [25], FastQ Groomer [26], Trim Galore [26], and FastQ Trimmer [27] tools. The data quality is assessed using the FASTQC tool, representing the base quality, GC content, duplicates, and adapter contaminations. Data is cleaned and prepared for spliced alignment. The detailed protocol is presented in Appendix A.

### 2.2. Alignment: Reference Genome

The pre-processed reads are mapped with the human genome (hg38) using splice-aware aligners. RNA-Seq reads should be mapped carefully because there is a chance that reads spanning splice junctions are incorrectly mapped. Handling alternative splicing in neurodegenerative disorders is especially important. STAR (Spliced Transcripts Alignment to a Reference) [28] is a spliced aligner that uses a 2-way approach (searching and stitching) to optimize the mapping of reads to the reference genome. In this study, the % of uniquely mapped reads ranges from 89 to 94% in HD and 77 to 94% for controls (Appendix A).

### 2.3. SNP Calling

Mapped alignments were subjected to PCR duplicate removal using Rmdup [29] to avoid misinterpreting an error to be a variant. Variant calling is performed on these files using SAM tools—Mpileup [27] and VarScan [30], and the genomic locations of the variants are obtained using ANNOVAR [31]. We filtered the variants using the following criteria (i) Phred score > 30, (ii) Recalibrate the alignment near the variants, (iii) Ts/Tv > 1.5, and (iv) Minimum read depth is 10.

The variants uniquely found in HD samples were filtered out from the variants of control samples. The majority of these variants are found to be in the non-coding regions, which encompasses regulatory motifs for transcription factors and miRNA binding.

### 2.4. Effect of Noncoding Variants

To obtain further information about the noncoding variants, we used an integrated tool called SNP nexus [32], which implements algorithms of different noncoding variant scoring tools such as CADD [33], GWAVA [34], fitCons [35], EIGEN [36], FATHMM [37], DeepSEA [38] and FunSeq2 [39]. We shortlisted the variants based on the cut-off used in each tool. Variants that affect transcription factor binding are also analyzed using SNP2TFBS [40] and Haploreg [41]. We compared identified variants with GWAS variants of other neurodegenerative disorders such as Alzheimer’s disease, Parkinson’s disease, and Amyotrophic Lateral Sclerosis [19,20,21,22].

### 2.5. Effect of Variants on miRNA Binding

We compared the identified variants with miRsnpscore [42], a repository of variants of miRNA gene regulation, and polymiRTS [43], a database of variants in experimentally valid seed regions and target sites. The common variants are filtered for further analysis. We also checked whether the miRNAs are reported in previous studies of HD. The miRDB [44] is used in finding the miRNA and gene target interactions that are associated with miRNAs. Later, the gene targets are compared with differentially expressed genes.

### 2.6. Effect of Variants on Regulatory and Epigenetic Modifications

We compared the variants with various XQTL studies available at QTLbase [45] that are associated with neurodegenerative diseases. We performed this comparison to elucidate the role of identified variants on multiple aspects like splicing, expression, methylation and histone acetylation.

### 2.7. Differential Gene/Transcript/Transcript Proportion Expression Analysis

Differential Transcript Expression (DTE) is the transcript level expression in which changes in expression are observed in at least one transcript of the genes between control and disease samples. Differential transcript Usage (DTU) is used to examine the differential transcript proportion (transcript composition of a gene).

To identify the transcript abundance, we used a salmon pseudo lightweight transcriptome-based aligner [46]. Salmon is an alignment-free tool that does its job in a 2-step process—indexing and quantification. It provides the count of overlapped reads with transcripts using HG38 transcriptome sequences. We used tximport to convert the transcript abundance to gene quantification. We used DESeq 2 [47] to identify differentially expressed genes. We filtered the DEG based on foldchange (|log_2_foldchange| > 1 and false discovery rate < 0.05). We categorized these genes based on the grade (grade 3/grade 4) of HD samples and compared them with already reported differentially expressed genes of HD from various literature sources and HumanMine [48]. We quantify transcript expression and proportions using DRIMSeq [49] and stageR [50] to understand transcript usage and expression in both HD and control samples.

### 2.8. Functional Interaction Network & Module Enrichment

We constructed tissue-specific functional interaction and co-expression networks among variant genes, differentially expressed genes, and transcription factors using Reactome FI [51], HumanBase [52], and coExpressDB [53]. To understand the enriched biological function, we performed functional enrichment using ClueGO [54], cluster profile [55], KEGG [56], and InterPro [57] database.

## 3. Results

### 3.1. Elucidating Interlinks between Variant Effect and Transcriptome Expression

The identified variants are majorly located in noncoding regions. To obtain further information about the effect of noncoding variants, we used an integrated tool called SNP nexus and compared variants with GWAS studies. The variants, which are not matching GWAS, are identified as novel variants. Further, we quantified gene/transcript expression from the BA9 data and compared the expression profiles with the overlapping variants. Interestingly, we found that most of the variant-associated genes are differentially expressed at the transcript level (Appendix A).

#### 3.1.1. Role of Variants in Epigenetic Modifications

Novel and GWAS-associated variants affect the regulatory functions and epigenetic changes. We compared the variants identified in the present study with various XQTL studies available at QTLbase that are associated with neurodegenerative diseases. We performed this comparison to elucidate the effect of variants in splicing, expression, methylation, and histone acetylation. In addition, we also found that these variant-associated genes are differentially expressed either at the transcript or gene level, as shown in Figure 1. Most of these variants may play an active role in histone modifications and associated expression change in grade 3 and grade 4, as reported in epitranscriptome studies, which also showed similar changes in the histone modification pattern in HD [16,17,18].

Table 1 depicts the effect of variants in epigenetic modification and associated expression patterns in gene and transcript levels in HD. We observed that these variants overlap with the transcript position (obtained with Ensembl [58]), and the transcripts are differentially expressed between control and HD subjects. We have compared our variants with already reported GWAS studies of neurodegenerative disorders, and the variants which are not reported in the GWAS studies are termed “novel.” QTL studies showed that the IL18 variant modulates histone methylation, expression and splicing. This variant overlapped with ENST00000280357.11, ENST00000524595.5, ENST00000525547.5 and ENST00000534225.1. We also obtained a similar differential transcript expression pattern. The transcript expression and proportion plots of the below-discussed genes are presented in Appendix A.

IL-18 is involved in neuroinflammation and abundantly expressed by microglia and inflammasome. Recent transgenic mouse studies showed that inhibiting inflammasome reduces IL-18 activation and thus provides neuroprotection [59]. rs360720 (A/G) variant overlapped with the four transcripts and affected their expression pattern in HD BA9 samples. The SLC9A5 gene is involved in Na+/H+ exchange, interacts with the AMPA receptor, and controls ion equilibrium during neurotransmission. Novel variant-associated genes (rs61740454 (G/C)) affect the transcript expression and upregulated HD. MPRIP gene participates in regulating the actin fiber. Studies showed that increased gene expression in neuronal cells affects the actin-binding and stress fiber assembly. MPRIP variants overlap with the transcripts (ENST00000334209.9, ENST00000417669.6, ENST00000579361.1) and modulate the expression.

The GAK1 gene plays a vital role in regulating Na+/K+ exchange and maintains the resting membrane potential [60]. The novel variant (rs17165089 (G/C)) affects the transcript expression. Epigenetic roadmap studies showed that it affects the histone modification pattern and is upregulated in HD. GOLGB1, located in the Golgi apparatus, is involved in protein metabolism and regulates the Golgi stress response. Epigenetic studies showed that a novel variant (rs77570895 (C/T)) influences methylation patterns and expression. In addition, this variant affected the transcript (ENST00000491690.1) expression and downregulated in HD. CYB5A is involved in electron transport and metabolism. PD-associated variant rs2032263 (G/C) disrupts the splicing, affecting the transcript expression (ENST00000299438.13, ENST00000397914.4) and also downregulated in HD conditions.

In summary, genes involved in ion homeostasis, stress response, and neuroinflammation are upregulated, and metabolism-related genes are downregulated in HD pathogenesis. In this study, we identified variants overlapping with the transcripts. Further, we found the interlink between the effect of variants in terms of transcript expression and epitranscriptomic modifications.

#### 3.1.2. Effect of Variants on Transcription Factor Binding and Histone Methylation/Acetylation Modification

The variants located in the noncoding region affect the regulatory elements responsible for transcription factor binding (TF), which indirectly affects the downstream gene expression. In Table 2, we listed some crucial variants and affected TFs that are differentially expressed and showed changes in the histone acetylation and methylation profile.

The gene LRRN4CL is responsible for the positive regulation of apoptosis and participates in the tyrosine kinase receptor signaling pathway. This gene is upregulated in HD conditions, and its expression pattern is not reported in the literature. Comparing this variant with QTL, studies showed this variant potentially affects expression and splicing patterns. This GWAS variant rs2512561 (G/A) disrupts the TF binding of NFKB1 and plays a bidirectional role in neuronal survival/death by providing neurotrophic factors and neuroinflammatory response. NFKB1 is downregulated in HD BA9 samples. The B3GNT8 gene present in the exosome membrane helps to transport proteins. PD-associated variant (rs284662 (C/T) affects the histone acetylation and methylation process, affecting the expression pattern. The B3GNT8 gene is upregulated in HD, and its expression pattern is not reported before in HD. The CPZ gene participates in WNT signaling, which plays a crucial role in cell survival/apoptosis. The CPZ novel variant rs13121547 (G/T) affects the histone acetylation pattern, which in turn upregulates the gene expression in HD, and this gene variant/expression is not reported so far in HD. In addition, B3GNT8 and CPZ variants affect the transcriptional repressor CTCF binding, which regulates the histone acetylation pattern. This gene is downregulated in HD, which may lead to the overactivation of the histone acetylation pattern. From our variant analysis, we showed most of the variants (77% and 68% variants in grade 3 and grade 4, respectively), are affecting the histone acetylation process (Figure 1).

SLC13A4 is involved in transporter activity, and the GWAS variant (rs3112355 (C/T)) affects the histone acetylation pattern, and this gene is upregulated in HD. This gene variant affects the TF binding of Prrx2 and is involved in vesicle trafficking; this gene is downregulated in BA9 HD, and PRRX2 expression is not explored in HD. PTGDR and PDGFD genes participate in vasodilation and cell migration. GWAS variants (rs4898758, rs11226059) affect the expression pattern, and the MAFF and RUNX1 TF binding is involved in cell proliferation and NOTCH signaling. These genes are upregulated in HD BA9 samples. The DSP gene is involved in the developmental process and the regulation of bundle of His cells to Purkinje myocyte communication. It disrupts the binding of the SOX10 transcription factor, which affects angiogenesis regulation. In this study, we elucidate the relationship between the variant-associated genes and their effect on epigenetic and transcription factor binding, which affects the downstream genes.

Interestingly, about 9.8% of the variants disrupting transcription factor binding affect the binding of novel TF Prrx2, whose binding is affected by 185 and 76 variants in grades 3 and 4, respectively. Prrx2 [61] is a novel gene that is upregulated in HD. It is a gene of paired-related homeobox 2 involved in ERK cascade, PI3K/AKT signaling, and WNT/ ß-catenin signaling pathway and is crucial for cell survival and migration. ERK activation [62] is observed in HD patients, affecting BDNF signaling, apoptosis, glutamate signaling, and EGF signaling. Aberration of PRRX2 may affect neuronal cell survival. In essence, the identified variants affect the transcription factor binding and their respective expression pattern through epigenetic modification.

#### 3.1.3. Effect of Variants on miRNA Binding and Its Expression

Noncoding variants potentially affect the binding motifs of miRNA. We compared our variants with miRsnpscore, a repository of variants affecting miRNAs gene regulation, and polymiRTS, a database of variants in experimentally valid seed regions and target sites. The common variants are filtered for further analysis and examined whether the miRNAs are reported in previous studies of HD. miRDB [44] is used to find the miRNA and gene target interactions. Later, the gene targets are compared with differentially expressed genes. The variants affecting the miRNA binding and their expression are shown in Table 3.

DRAM1 is downregulated in HD. The FARP1 gene participates in synaptic retrograde transport, and the GWAS variant (rs2282048) affects the miR-100 binding and its expression. CSNK1A1 variant modulates the binding of miR-374a, and this miRNA is upregulated in HD, and target gene expression is downregulated. This gene is involved in WNT signaling and is a potential biomarker for Alzheimer’s disease. TMEM43 is involved in cardiovascular disease and plays a major role in metabolic pathways; this gene variant changes the binding of miR-30a and is upregulated in HD, and TMEM43 is downregulated in HD BA9 samples. Furthermore, we found other potential target genes for the above-discussed upregulated miRNA. We noticed that all downstream target genes for these miRNAs were also downregulated in HD.

### 3.2. Differential Gene Expression/Transcript Expression

We performed transcriptome-based quantification using salmon. In grade 4, we found 33 differentially expressed genes reported in HD and 38 novel genes in our study. In grade 3, we found eight differentially expressed genes already reported in HD and 27 novel genes. Common genes of grade 3 and grade 4 are 59, of which 23 are novel, and 36 are already reported in HD.

#### 3.2.1. Grade-3 Specific Gene Expression

In grade 3, most of the mitochondrial-related transcripts are downregulated (MT-TS1, MT-TF, MT-TL2, MT-TW, MT-TD, MT-TG, MT-TR, MT-TV, MT-TL1, MT-TK, MT-TM, MT-TH and MT-TN) and these gene expressions are not reported in HD. INSRR gene is involved in the ERK cell surviving signaling pathway and is upregulated in grade 3-HD. We found that the PITX2 gene is important for GABA interneuron development and is upregulated in HD, as reported previously. The HOXA10 gene participates in histone deacetylation and is found to be upregulated in grade 3. CXCR2 and B2M genes play a crucial role in immune response, and this gene is upregulated in grade-3 HD. We have listed all the differentially expressed genes of grade 3 in Appendix A. In summary, mitochondrial-related genes are downregulated 3, and immune response genes are upregulated in grade 3. This may lead to energy dysfunction and associated neuroinflammation at the early stages of the disease. The normalized TPM count plots for some of the above-discussed genes are represented in Figure 2.

#### 3.2.2. Grade-4 Specific Gene Expression

In grade 4, we found that the TBX15, WNT6 and SFRP2 genes are related to cell fate commitment, and WNT signaling is upregulated and participates in apoptosis. A novel gene, SFRP2, is reported to be negatively regulating the WNT signaling pathway [63], which is crucial for blood vessel formation and essential for vasculature maintenance [64]. OSR1 gene is involved in smooth muscle and pericyte regulation, which are important for blood pressure and vasculature morphogenesis.

CCR2 and GPR183 are involved in astrocyte migration, and these genes are upregulated in HD. The CD177, H19, IGHG3, IGKV3-20 and IGLC2 genes are involved in the inflammatory response, and these genes are upregulated in HD. USP17L2 gene is involved in MAPK and cell surviving signaling, which is downregulated in HD. NPHS1 plays a crucial role in actin filament and mediated transport, and this gene is downregulated in HD. MIR149 is involved in angiogenesis, and vasculature downregulates in HD. The above-discussed gene expression is not reported earlier in the literature. We have listed all the differentially expressed genes of grade 4 in Appendix A.

In both grades, the MMP9, HOXD10, BARX1, HOXA13 and CPZ genes are upregulated and participate in various functions such as regulation of cytochrome C regulation/apoptosis, motor neuron cell fate regulation and endothelial cell migration, and WNT signaling, respectively. The normalized TPM count plots for some of the above-discussed genes are shown in Figure 3.

#### 3.2.3. Comparison of Gene Expression (BA9/HD) with Literature Bulk RNA-Seq

To validate the expression pattern, we also compared the identified gene expression profiles of BA9 HD samples with other published bulk RNA seq studies in HD and the results are shown in Figure 4.

It is crucial to know that the genes in HD samples are exhibiting differential expression as in other reported bulk RNA Seq studies. This helps to confirm that the up/down-regulated genes of our study are also expressed in other studies. Most of the gene expression profiles are matched with the reported studies.

#### 3.2.4. Comparison of Gene Expression (BA9/HD) with GTEX (BA9) and Other Tissue

We compared the novel gene expression pattern in HD with huge control samples from BA9 and various tissues (GTEX consortium data) to examine whether the disease sample gene expression differs from the control tissues. The comparison of novel gene expression patterns with GTEX and other tissue control samples is shown in Figure 5.

It is observed that the gene expression pattern of HD differs from that of the controls and also normal samples from other brain tissues. We found that the control BA9 TPM values match the GTEX BA9 and exhibit similar expression patterns. The novel gene expression in HD differed from that of the other control expression, which shows that these genes are potentially dysregulated in HD BA9.

### 3.3. Network and Tissue-Specific Functional Enrichment

#### 3.3.1. Gene Co-Expression Network and Function Module Network for Differentially Expressed Genes

We have constructed a co-expression network (Figure 6) for the differentially expressed genes using COEXPRESSdb [53]. Genes that are co-expressed together tend to have similar functions. Here we discuss some of the DEGs based on the network properties (degree and betweenness centrality). MRC1 is a novel HD gene co-expressed with CD163, VSIG4, and MS4A4A, and these genes are involved in vasculature regulation. ANXA2 is an upregulated gene co-expressed with S100A4 and S100A6 genes, and these genes are active participants in interleukin signaling that promotes neuroinflammation in HD [65]. GZMK is also co-expressed with PTGDR, which has a major role in MAPK cascade and immune response. GZMK and PTGDR are upregulated, and their involvement in MAPK signaling can be considered as active participation in cell survival mechanisms [66]. OSR1 gene is involved in pericyte migration and maintains the vasculature, and this gene co-expressed with the LRRN4CL, FOXF2 and OSR2 genes. Novel TF FOXD2 is co-expressed with FOXD1 and WNT6.

#### 3.3.2. Tissue-Specific Function-Interaction Network

We constructed a functional interaction network using novel differentially expressed genes and transcription factors (Figure 7) and associated downstream gene expression. We have used ReactomefiVIZ [51] to construct a functional interaction network. This tool uses interactions obtained from pathways in the KEGG and Reactome knowledgebase and merges them with predicted interactions using a machine learning approach to annotate function interactions.

Figure 7 shows that TCF12 plays a significant role in cell fate determination. This TF binding was affected by EP300 and FOXD1 novel variants. Novel DEGs such as HOXA11, HOXD8, HOXD9, NKX2-5, HOXD10, and FOXD1 genes are activated by TCF12. All these downstream genes are dysregulated in HD. We predict that novel differentially expressed genes KRT19 and TBX15 may be activated by ESR1 and EP300, respectively, based on the information obtained from function interactions using the ReactomeFI tool [51]. EP300 also regulates the expression of TCF12, an upregulated transcription factor in HD. EP300 is a variant-associated gene known for its function in histone acetylation [67] and plays a significant role in epigenetic modifications and gene expression [68].

These interaction networks revealed the functional relationship between DEGs and TFs. Variant genes are also indirectly affecting these functional links as they alter the binding of transcription factors that are majorly involved in the histone acetylation process. Modulation in the TF network affects the downstream target genes.

#### 3.3.3. Tissue-Specific Functional Module Network

Differentially expressed genes were subjected to tissue-specific functional module enrichment analysis. Modules represent different functions in which the genes are involved. The network of tissue-specific function modules for BA9 is shown in Figure 8. These modules consist of genes responsible for angiogenesis, vasculature development, epithelial cell differentiation, immune response, and G-protein coupled receptor signaling pathway. Most of the genes involved in the vasculature and blood vessel-related functions are dysregulated in HD pathogenesis. This study showed that most of the genes related to immune response are differentially expressed, which may lead to aberrant neuronal inflammation. We also found that genes involved in vasculature maintenance are dysregulated, which may affect the blood-brain integrity and add stress burden leading to inflammation.

### 3.4. Proposed Mechanism for Neurodegeneration

Neuroinflammation plays a bifacial role in cell survival and death. Massive amounts of neuroinflammatory cytokines released from hypoxic/injured neurons activate the inflammatory microglia and reactive astrocytes (Figure 9). By analyzing the microenvironment, microglia and astrocytes provide trophic or cytotoxic signals. This massive cytokine rush affects the endothelial structure and distorts tight junctions and pericytes in blood vessels. Vascular abnormalities affect smooth muscle control that eventually strikes vasodilation and constriction. Loss of coordination between neurons, blood vessels and astrocytes hampers the energy metabolism process. From the transcriptome and functional enrichment analysis, we propose that aberration in the vascular region and neuroinflammation affects the structural integrity, which hampers the metabolic pathways, making the associated neurons vulnerable in HD.

### 3.5. Limitations

This limitation of the study is the small size of data (population of HD samples) obtained from BA9. Although we have compared the quantified HD expression with GTEX and other bulk RNA seq data, we still need a large population to understand the variant effects in expression patterns.

## 4. Conclusions

In this study, we explored the effect of variants on transcript expression and splicing patterns. Most of the identified variants are located in the noncoding region and modify the regulatory elements. Novel variants associated with the following genes, GAK1, SLC9A5, MPRIP, GOLGB1 and GAK1, affect the histone acetylation pattern. However, these variants are overlapped with their transcript and are upregulated in HD at both gene and transcript levels. Seven novel variants impact the transcription factor binding of HOXA5, FOXD1 and PRRX2 and play a crucial role in homeostasis and neuromuscular signaling pathways. These TFs are dysregulated in HD and have not been reported so far. Interestingly, we found a novel variant in the CPZ gene that affects the histone acetylation pattern and is upregulated in HD. The CPZ variant modulates the CTCF binding and is a well-known histone acetylation repressor. The CTCF gene is downregulated in HD. We also found the variant genes affecting miRNAs and their downstream target genes. Identified variants affect the miRNA binding. These miRNAs are upregulated in HD, and their downstream target genes are downregulated in HD.

Differential gene expression analysis showed that mitochondrial function-related genes and inflammatory-responsive genes are dysregulated at grade 3. Blood vessel morphogenesis, cell survival signaling, and angiogenesis genes are downregulated in grade 4; Inflammatory related genes are tremendously upregulated in grade 4. 33 genes of grade 4 differentially expressed genes are upregulated in order to activate Aryl hydrocarbon receptor (Ahr) transcription factor that is responsible for the degeneration of motor neurons and behavioral activity related neurons. The removal of Ahr is beneficial and gives a neuroprotective effect to the HD samples, as previously reported [69].

We explored the BA9-specific function interaction networks to understand the relationship between novel differentially expressed genes and transcription factors. Tissue-specific functional enrichment showed that predominant genes related to vascular architecture are dysregulated in HD. From this detailed analysis, we propose that abnormal neuroinflammation affects endothelial cell integrity, affecting BBB homeostasis.

## Figures and Tables

**Figure 1 genes-13-02385-f001:**
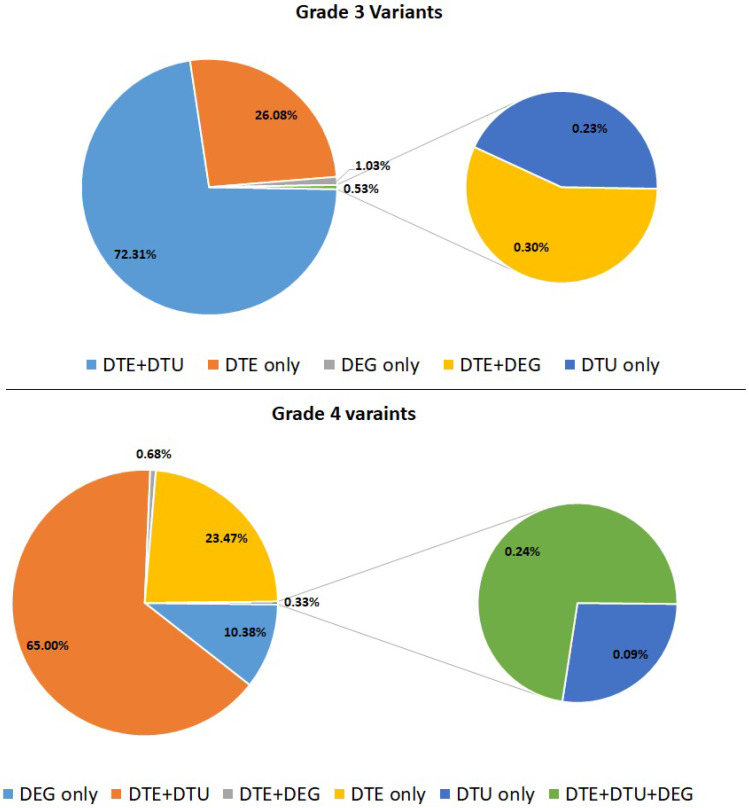
Variants involved in epigenetic modifications in differential transcript/gene expression profiles. DEG only: Variant genes that are differentially expressed in the study; DTE: variant genes which are overlapped with genes of differential transcript expression; DTU: variant genes overlapped with Differential transcript usage.

**Figure 2 genes-13-02385-f002:**
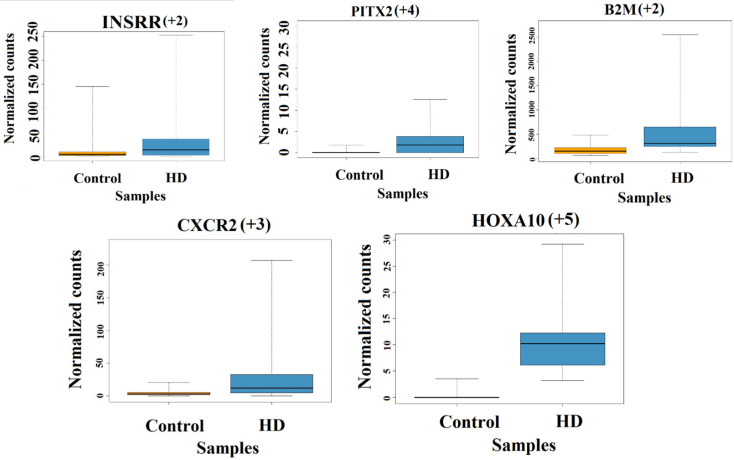
Box plot showing differentially expressed genes present in grade 3 HD samples. The box and the line inside the box represent the interquartile interval of the data (normalized counts) and median, respectively. The whiskers/lines connecting the box on both sides towards the top and bottom are the maximum and minimum values in the data, respectively. The adjusted *p*-value (p_adj_) is less than 0.05; The + sign denotes upregulation. The numbers in parenthesis denote the foldchange.

**Figure 3 genes-13-02385-f003:**
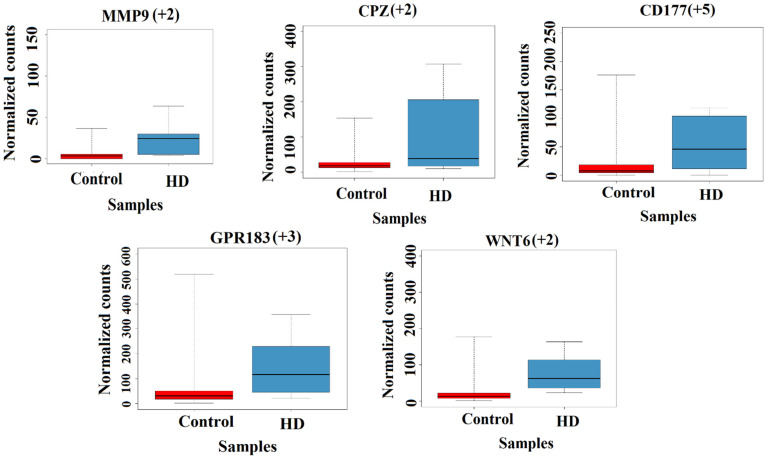
Box plot showing differentially expressed genes present in grade 4 HD samples. The box and the line inside the box represent the interquartile interval of the data (normalized counts) and median, respectively. The whiskers/lines connecting the box on both sides towards the top and bottom are the maximum and minimum values in the data, respectively. The adjusted *p*-value (p_adj_) is less than 0.05; The + sign denotes upregulation. The numbers in parenthesis denote the foldchange.

**Figure 4 genes-13-02385-f004:**
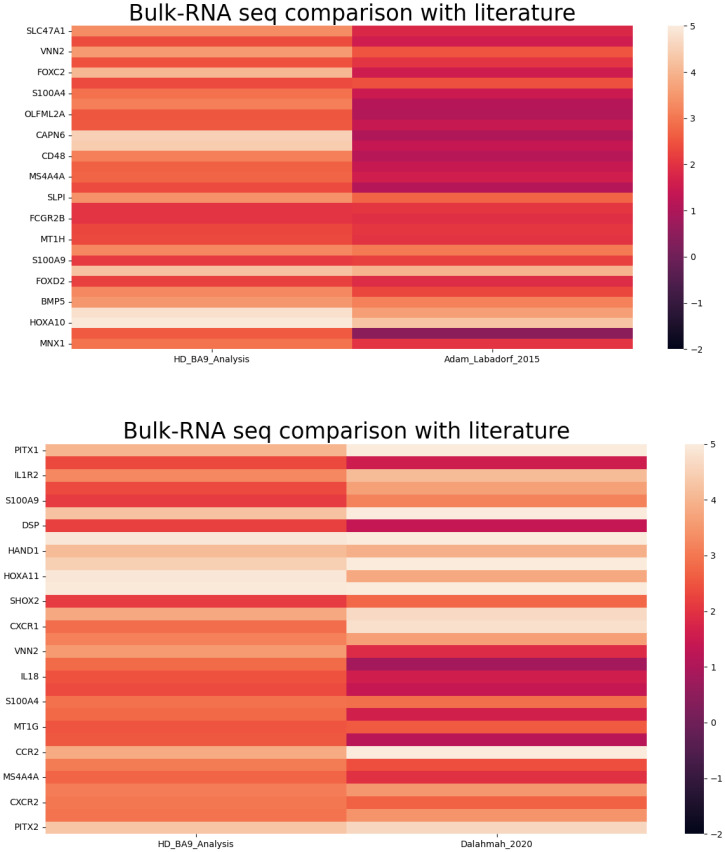
Comparison of gene expression profiles of HD—BA9 RNA-seq with bulk RNA Seq expression data.

**Figure 5 genes-13-02385-f005:**
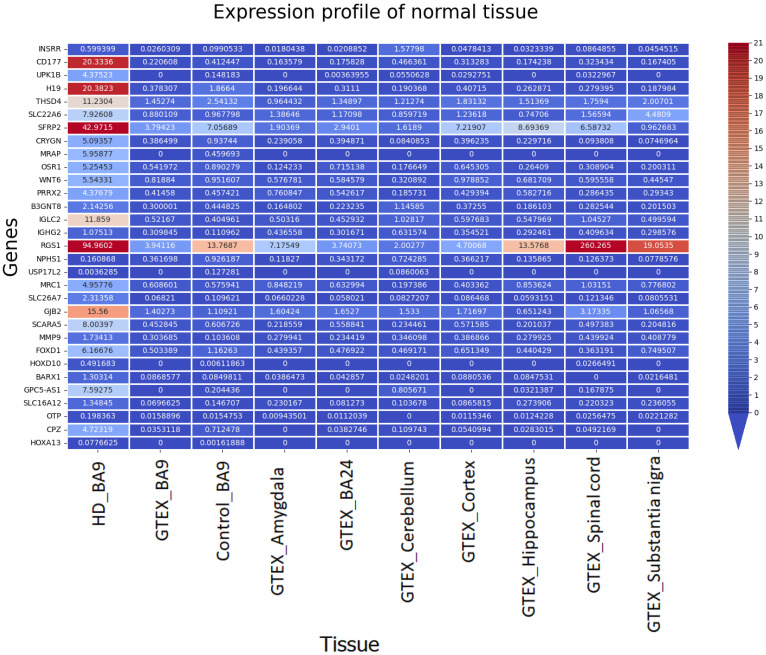
Comparison of TPM count (HD) with BA9 and other GTEX brain tissue.

**Figure 6 genes-13-02385-f006:**
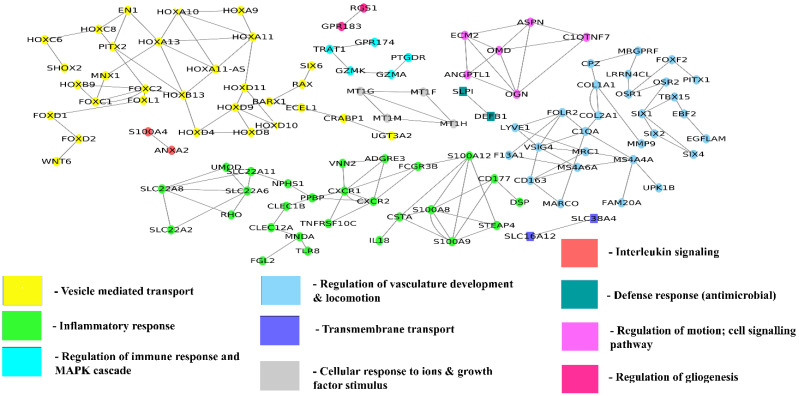
Gene co-expression network of differentially expressed genes of grade 3 and grade 4. The node color in the network represents the function in which most of the genes are enriched.

**Figure 7 genes-13-02385-f007:**
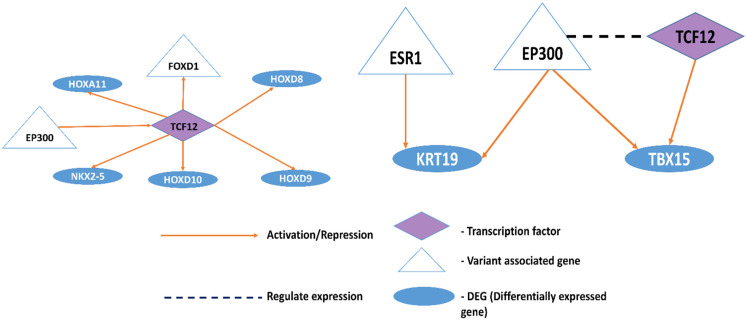
Function interaction network of differentially expressed genes, variant genes, and transcription factors.

**Figure 8 genes-13-02385-f008:**
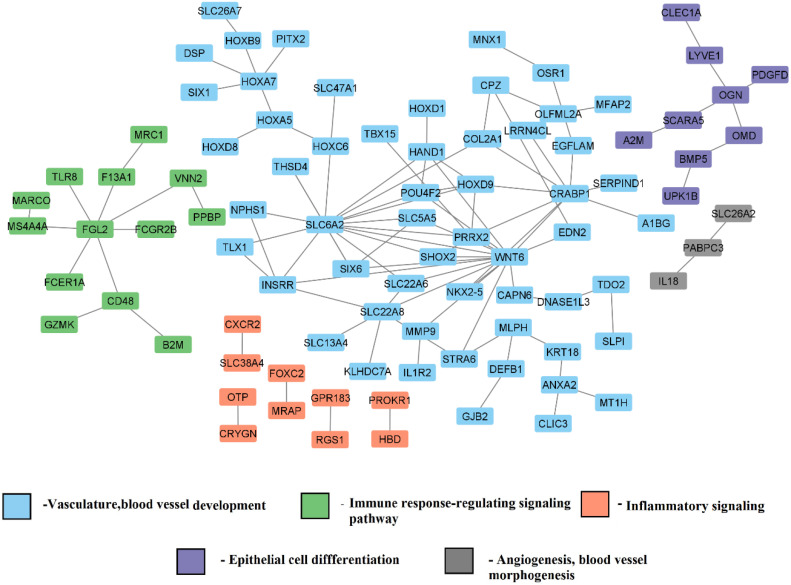
Tissue-specific functional module network of differentially expressed genes (grade 3/grade 4).

**Figure 9 genes-13-02385-f009:**
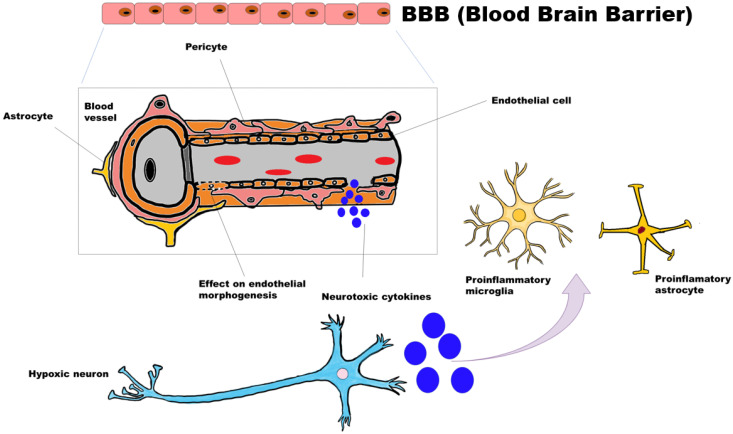
Proposed mechanism for neurodegeneration in HD.

**Table 1 genes-13-02385-t001:** Effect of variants in transcript expression profiles and epigenetic modification.

Chr	REF/ALT	Gene	log_2_FC	Variant	Transcripts	Novel/Reported	QTL Category	Expression
Chr11	A/G	IL18 (Grades 4 & 3)	2.484262691	rs360720	ENST00000280357.11, ENST00000524595.5, ENST00000525547.5, ENST00000534225.1	PD, AD	hQTL, eQTL, mQTL, sQTL	UP
chr11	G/A	USH1C (Grades 4 & 3)	11.8	rs2041027	ENST00000529563.5	PD, AD	hQTL	UP
Chr16	G/C	SLC9A5 (Grades 4 & 3)	11.1	rs61740454	ENST00000564704.5	Novel	hQTL	UP
Chr17	C/A	MPRIP (Grades 4 & 3)	5.1	rs3826304	ENST00000579361.1	Novel	hQTL	UP
Chr22	G/A	MPPED1 (Grades 4 & 3)	4.9	rs739018	ENST00000334209.9, ENST00000417669.6	PD	hQTL, Eqtl	UP
Chr4	G/A	GAK (Grades 4 & 3)	3.2	rs17165089	ENST00000504668.5	Novel	hQTL	UP
Chr7	C/T	SLC13A4 (Grades 4 & 3)	2.588	rs3112355	ENST00000471405.5, ENST00000480376.1	PD, AD	eQTL	UP
Chr3	C/T	GOLGB1 (Grades 4 & 3)	−5.4	rs77570895	ENST00000491690.1	Novel	eQTL, mQTL	DOWN
Chr18	G/A	CYB5A (Grades 4 & 3)	−7.6	rs2032263	ENST00000299438.13, ENST00000397914.4	PD	eQTL, sQTL	DOWN
Chr15	T/C	ADAMTSL3 (Grades 4)	NA	rs1480822	ENST00000569510.5	PD	hQTL, eQTL	Novel

**Table 2 genes-13-02385-t002:** Effect of variants on transcription factor binding.

CHR	REF/ALT	GENE	Variant Gene Expression	Variant Gene Expression Pattern	SNP	Category (Methylation, Acetylation, Gene Expression, Splicing)	Transcription Factor Affected	Transcription Factor Expression	GWAS Study
chr11	G/A	LRRN4CL	Novel	UP	rs2512561	eqtl, sqtl, mqtl	NFKB1	DOWN	PD, AD
chr19	T/C	B3GNT8	Novel	UP	rs284662	hqtl, eqtl, sqtl, mqtl	CTCF	DOWN	PD
chr4	G/T	CPZ	Novel	UP	rs13121547	Hqtl	CTCF	DOWN	Novel
chr7	C/T	SLC13A4	Reported	UP	rs3112355	Eqtl	Prrx2	UP	PD, AD
chr11	C/G	PDGFD	Reported	UP	rs11226059	Eqtl	MAFF	UP	AD
chr12	C/T	COL2A1	Reported	UP	rs2070739	hqtl, sqtl	Tcf3	DOWN	PD, AD
chr14	A/G	PTGDR	Reported	UP	rs4898758	hqtl, eqtl	RUNX1	UP	PD
chr14	C/G	PDGFD	Reported	DOWN	rs813921	Hqtl	Pdx1	DOWN	Novel
chr6	G/A	DSP	Reported	UP	rs6929069	Eqtl	SOX10	UP	PD

**Table 3 genes-13-02385-t003:** Variants that affect miRNA binding and its expression in HD.

Chr	Alt/Ref	Gene	SNP	GWAS/Novel SNP	Affected miRNA	miRNA Expression	miRNA Target Gene/s	Expression of Gene
Chr12	C/T	DRAM1 (DOWN)	rs17032062	PD, AD	miR-302e, miR-101	Up	SLC16A12, SCARA5, HOXD8, PTGDR, FGL2, FCGR2B, SLC26A2, DSC3, SLC26A7, OGN	DOWN
Chr13	A/G	FARP1 (DOWN)	rs2282048	PD, AD	miR-100	Up	SIX1	DOWN
Chr5	T/A	CSNK1A1 (DOWN)	rs11167469	PD	miR-374a	Up	CD177, CAPN6, OGN, HOXA10, SLC26A2, RGS1, MS4A4A, FOXD2,	DOWN
Chr3	G/T	TMEM43 (DOWN)	rs3796308	PD, AD	miR-30a	Up	HOXA13, CLEC12A, MRC1, SIX1, DSP, FOXD1, SCARA5, SPTLC3, RGS1	DOWN

## Data Availability

Data will be made available upon request.

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
