# Peer review of "Integrative Meta-Analysis of Huntington’s Disease Transcriptome Landscape"

_genes, 2022, doi:10.3390/genes13122385_

Round 1

Reviewer 1 Report

 Sneha et al present an interesting manuscript on “multi-omics analysis of Huntington's disease transcriptome landscape”. The manuscript has well-designed experiments and claims are mostly based on experimental evidence. However, before publication, addressing the following comments will improve the manuscript to the level of publication:

1. In the abstract or at the end of the introduction, authors should mention from where the sample/data is taken

2. All Figures legend should be more comprehensive with details of each panel & graph, otherwise difficult to understand.

3. Statistics is required in the graph of Figures 2 and 3. What does the error bar represent?

4.    In Supplementary Fig S2, the graph can be divided into three parts with details in the figure legends for each part. Moreover, DTE/DTU full form is required in the figure legend.

5.    In line 181: direct experimental evidence missing for the claim that variants induce gene expression change via histone modifications. Either author should provide experimental evidence or tone down their claim.

6.    Line 392: “Novel differentially expressed genes KRT19 and TBX15 are activated by ESR1 and EP300 respectively”

Not clear if this is the author’s claim or mentioned in published literature. Direct experimental evidence is required if it is the author’s claim otherwise please give a citation here.

Minor points:

1.    Authors should define DTU/DTE in the manuscript.

2.    In the supplementary file, the author’s address is mentioned as “Institute of Technology Madras” which is different from that mentioned in the main manuscript.

Author Response

We thank the reviewer for their constructive comments. We have carefully addressed all the points suggested by the reviewers, and the corrections are incorporated in the revised manuscript.

Reviewer 1

Sneha et al present an interesting manuscript on “multi-omics analysis of Huntington's disease transcriptome landscape”. The manuscript has well-designed experiments and claims are mostly based on experimental evidence. However, before publication, addressing the following comments will improve the manuscript to the level of publication:

Response: Thanks for the constructive comments. We have incorporated all the necessary corrections in the revised manuscript.

  1. In the abstract or at the end of the introduction, authors should mention from where the sample/data is taken

Response: We have obtained the samples from SRA database. This information has been included in the revised manuscript (Abstract).

  1. All Figures legend should be more comprehensive with details of each panel & graph, otherwise difficult to understand.

Response: We have included the details in the legends of each figure.

  1. Statistics is required in the graph of Figures 2 and 3. What does the error bar represent?

Response: We have carried out statistical tests and the padj value is less than 0.05. This information has been included in the footnote of Figures 2 and 3.

Figures 2 and 3 are box plots for normalized counts obtained from HD and control samples. Box and line inside the box represent, interquartile interval of the data (normalized counts) and median, respectively. The whiskers/lines connecting the box both sides towards top and bottom are maximum and minimum values in the data respectively.

  1. In Supplementary Fig S2, the graph can be divided into three parts with details in the figure legends for each part. Moreover, DTE/DTU full form is required in the figure legend.

Response: We have divided supplementary figures into three parts and provided legends appropriately. The expansion for DTE/DTU is included in the footnote and methods section.

  1. In line 181: direct experimental evidence missing for the claim that variants induce gene expression change via histone modifications. Either author should provide experimental evidence or tone down their claim.

Response: Epitranscriptomic studies showed that changes in histone modifications are predominant in HD and eventually affect the downstream expression. We have included the reference and modified the sentence (Section 3.1.1).

Question 6: “Novel differentially expressed genes KRT19 and TBX15 are activated by ESR1 and EP300 respectively” Not clear if this is the author’s claim or mentioned in published literature. Direct experimental evidence is required if it is the author’s claim otherwise please give a citation here.

Response: We observed this information from the functional network constructed using ReactomeFI tool, which utilized various pathways such as KEGG, regulatory etc. We have included the reference and modified the sentence appropriately (Section 3.3.2).

Minor points:

  1. Authors should define DTU/DTE in the manuscript.

Response: We have included the expansion DTE/DTU in methods section.
(Section 2.7)

  1. In the supplementary file, the author’s address is mentioned as “Institute of Technology Madras” which is different from that mentioned in the main manuscript.

Response:  We have corrected the address in the main manuscript and supplementary information.

Reviewer 2 Report

In the current manuscript, entitled “Integrative Multi-omics analysis of Huntington's disease transcriptome landscape”, Sneha et al., investigated the relationship between variants and their effect on gene/downstream transcript expression in HD. They computed gene and transcript abundance and identified variants from RNA seq. Further, they predicted novel variants by several noncoding variant scoring tools. Next, they showed many variants affect the histone acetylation pattern miRNA binding in HD. Finally, they claimed abnormal neuroinflammation acts as a two-edged sword that indirectly affects the vasculature and associated energy metabolism.

Generally, this paper was not organized well and clearly presented. It is not accurate to say multi-omics in the title by only doing RNAseq. The authors do not provide discussion to address the advantage and progress made by other researchers in this filed. It is not adequate to be published as an incomplete manuscript.

Some suggestions:

For all figures, the authors should provide detailed legend to depict the figure. I have hard time to follow the author’s points. The font is too small to read.

Fig3 and Fig4 are both about grade 4. They may be combined to make the manuscript clearer. They may need to validate their novel variants (at least some of them) in HD tissue to make their paper more solid.

All tables are not integrated well in the manuscript. 

Author Response

We thank the reviewer for their constructive comments. We have carefully addressed all the points suggested by the reviewers, and the corrections are incorporated in the revised manuscript.

Reviewer 2

In the current manuscript, entitled “Integrative Multi-omics analysis of Huntington's disease transcriptome landscape”, Sneha et al., investigated the relationship between variants and their effect on gene/downstream transcript expression in HD. They computed gene and transcript abundance and identified variants from RNA seq. Further, they predicted novel variants by several noncoding variant scoring tools. Next, they showed many variants affect the histone acetylation pattern miRNA binding in HD. Finally, they claimed abnormal neuroinflammation acts as a two-edged sword that indirectly affects the vasculature and associated energy metabolism.

Response: Thanks for the constructive comments. We have incorporated all the necessary corrections in the revised manuscript.

  1. Generally, this paper was not organized well and clearly presented.

Response: We have edited the manuscript appropriately and presented the results clearly.

  1. It is not accurate to say multi-omics in the title by only doing RNAseq. 

Response: Thanks for the comment. We have changed the title to “Integrative Meta-analysis of Huntington's disease transcriptome landscape”

  1. The authors do not provide discussion to address the advantage and progress made by other researchers in this filed. It is not adequate to be published as an incomplete manuscript.

Response: We have provided the discussion to address the advantage and progress made by other researchers in this field and added the necessary references in the introduction section (Section 1). Further, we have provided related references and discussed in the results section and conclusion (Section 3.1.1, Section 3.3.1, Section 4)

  1. Fig 3 and Fig 4 are both about grade 4. They may be combined to make the manuscript clearer.

Response: We have combined the information available in Figures 3 and 4, and highlighted the important gene expressions in Figure 3.

  1. They may need to validate their novel variants (at least some of them) in HD tissue to make their paper more solid.

Response: We have included the experimental data available in the literature to support our observations and validate the novel variants. Our lab is computational and carrying out experiments is beyond the scope of the present study.

Round 2

Reviewer 2 Report

The authors fully addressed most of my questions. I only have minor point.

Please check font of Fig6, it looks smaller than Fig8.